# A mathematical model to quantify the effects of platelet count, shear rate, and injury size on the initiation of blood coagulation under venous flow conditions

**Anass Bouchnita**[1][☯]*, **Kirill Terekhov**[2][☯], **Patrice Nony**[3], **Yuri Vassilevski**[2,4,5], **Vitaly Volpert**[2,6,7,8]

**1** Ecole Centrale Casablanca, Casablanca, Morocco, **2** Marchuk Institute of Numerical Mathematics, Russian Academy of Sciences, Moscow, Russia, **3** Services de Pharmacologie Clinique, Hospices Civils de Lyon, Lyon, France, **4** Sechenov University, Moscow, Russia, **5** Moscow Institute of Physics and Technology, Dolgoprudny, Russia, **6** Institut Camille Jordan, Université Lyon 1, Villeurbanne, France, **7** INRIA team Dracula, INRIA Lyon La Doua, Villeurbanne, France, **8** Peoples' Friendship University of Russia (RUDN University), Moscow, Russia

☯ These authors contributed equally to this work.
* anass.bouchnita@centrale-casablanca.ma

**Data Availability Statement:** The entire set of equations, boundary conditions and parameter values used is presented in the article. These data

## Abstract

Platelets upregulate the generation of thrombin and reinforce the fibrin clot which increases the incidence risk of venous thromboembolism (VTE). However, the role of platelets in the pathogenesis of venous cardiovascular diseases remains hard to quantify. An experimentally validated model of thrombin generation dynamics is formulated. The model predicts that a high platelet count increases the peak value of generated thrombin as well as the endogenous thrombin potential (ETP) as reported in experimental data. To investigate the effects of platelets density, shear rate, and wound size on the initiation of blood coagulation, we calibrate a previously developed model of venous thrombus formation and implement it in 3D using a novel cell-centered finite-volume solver. We conduct numerical simulations to reproduce *in vitro* experiments of blood coagulation in microfluidic capillaries. Then, we derive a reduced one-equation model of thrombin distribution from the previous model under simplifying hypotheses and we use it to determine the conditions of clotting initiation on the platelet count, the shear rate, and the plasma composition. The initiation of clotting also exhibits a threshold response to the size of the wounded region in good agreement with the reported experimental findings.

## 1 Introduction

Blood coagulation can be initiated when the tissue factor (TF) is exposed to the bloodstream [1]. In arteries, platelets initiate thrombus formation by adhering to the subendothelium and recruiting other platelets. As a result, we refer to the thrombi developed in arteries by 'white clots' because they consist mainly of platelets. In venous flows, fibrin propagates during the

are sufficient to reproduce the results presented in the article.

**Funding:** K. Terekhov was supported by RFBR grant 18-31-20048. Yu. Vassilevski was supported by the world-class research center "Moscow Center for Fundamental and Applied Mathematics" (agreement with the Ministry of Education and Science of the Russian Federation No. 075-15-2019-1624). V. Volpert was supported by the "RUDN University Program 5-100." The development of the 3D numerical model of thrombus growth under venous flow conditions was supported by Russian Science Foundation grant 14-31-00024. The funders had no role in study design, data collection and analysis, decision to publish, or preparation of the manuscript.

**Competing interests:** The authors have declared that no competing interests exist.

early stages and prevents the platelets from aggregating. The formed thrombi are called 'red clots' because they are formed mainly by the fibrin mesh and RBCs [2]. In this case, platelets accelerate the pro duction of the fibrin mesh by expressing negatively charged phospholipids up on their activation. Studies, quantifying the relationship between platelets and the risk of venous thrombosis, have not attracted a lot of attention, because platelets play a more important role in arterial thrombus formation. Still, an elevated platelet count is considered to be a risk factor for the incidence of venous thromboembolism (VTE). In fact, it was shown that a platelet count higher than $350 \times 10^9 \, L^{-1}$ can be considered as an independent risk factor for VTE [3] and that it is also associated with the hypercoagulability of blood [4]. These clinical studies were confirmed by *in vitro* assays such as thrombin generation which measure the coagulability of blood samples from individual patients. In these assays, it was observed that both the peak thrombin and the endogenous thrombin potential (ETP) values increase as the density of platelets in plasma grows [5]. During thrombus formation, platelets are recruited into the thrombus and further increase its hydraulic resistance [6]. Hence, the role of platelets in the formation of venous thrombus consists in increasing the coagulability of blood and reducing the porosity of the thrombus. Quantitative studies clarifying the relationships between platelets, blood flow, as well as the other elements of hemostasis, are of paramount importance to develop better treatment strategies for VTE.

Blood coagulation is a complex process that leads to thrombus formation inside blood vessels. It can be initiated when the endothelial tissue is damaged and tissue factor (TF) is exposed to bloodstream. In this early phase of hemostasis, platelets adhere to exposed collagen and express TF as well as other clotting factors upon their activation [7]. TF form a complex with FVII and FVIIa that activates the blood clotting factors FIX and FX near the damaged tissues [8]. Factor FXa convert prothrombin (FII) into thrombin (FIIa) in the plasma which results in the generation of thrombin, the most important factor in the coagulation cascade. Thrombin converts fibrinogen into fibrin which subsequently forms the fibrin clot [9]. In the early stages of thrombus formation, blood clotting is only initiated when a sufficient amount of generated thrombin is able to trigger the amplification phase of the coagulation cascade. During this phase, thrombin activates the clotting factors FV, FVIII, and FXI. The latter activates FIX which converts FX into its active form. At the same time, FVIIIa forms a complex with FIXa that activates FX. Then, FXa and FVa forms a complex called prothrombinase which activates prothrombin. This results in a self-amplifying cycle of thrombin generation. This cycle can only be stopped by few mechanisms such as the inhibition of thrombin production by anti-thrombin and activated protein C (APC) as well as its mechanical removal by blood flow. Aggregated platelets reduce the velocity of blood flow and produce thrombin as well as other clotting factors which accelerate the coagulation process.

Various types of studies were conducted in order to understand the underlying mechanisms that lead to the initiation of thrombus growth under the flow. Theoretical investigations begin already during the 19th century with the Virchow's triad that divides the factors leading to thrombosis into three categories: blood stasis, hypercoagulability, and alterations affecting the vessel wall [10]. Experimental studies not only validated these hypotheses but also explored the relationship between the different factors involved in venous thrombosis pathogenesis. In an important work that falls into this category [11], *in vitro* experiments were conducted using microfluidic capillaries. A patch containing TF was inserted inside a tube with rigid walls where blood was flowing due to the action of a syringe pump. Initiation of coagulation time was measured for normal pooled plasma (NPP) and platelet-rich plasma (PRP) under different values of shear rate and for different sizes of the TF patch. The work is of particular importance because it provides quantitative results that show the existence of a threshold response for the initiation of the clotting process under different conditions.

There have been several methods that were used in the mathematical modelling of the kinetics of blood coagulation and the dynamics of thrombus growth [12]. The complexiy of each of the developed models depend on the extend of the scope it intends to achieve [13]. In order to evaluate the coagulability of blood samples from individual patients, researchers typically use deterministic models consisting of ordinary differential equations (ODEs) describing the concentrations of clotting factors over time [14–17]. The densities of platelet subtypes (activated and inactivated) can also be introduced as ODEs in these models. The advantage of this approach is the ability to compare the modelling results (thrombin generation curves) with the output of major coagulation assays such as the thrombin generation test. It is also possible to analyse these models mathematically to understand the dynamics of blood coagulation. Furthermore, these works can be used as a basis to develop more complex models that describe the spatial dynamics of thrombus growth by adding advection and diffusion terms to models in order to capture the spatial distribution of clotting factors. The kinematics of blood flow can be captured using continuous models such as the Navier-Stokes equations [18] or discrete methods such as Dissipative Particle Dynamics [19, 20]. The discrete approach is more appropriate to simulate the dynamics of platelet aggregation as observed in arterial cardiovascular events [2]. In order to describe the interaction between thrombus growth and hemodynamics, some studies consider clot as a permeable medium whose hydraulic resistance depends on the concentration of fibrin polymer [21], or the density of platelets [22]. Another approach consists in the modelling of the effects of the clot on the viscoelastic properties of blood flow [23]. In general, the models describing the formation of venous thrombi focus on the distribution of clotting factors in plasma while those studying the development of arterial white thrombi are primarily interested by the dynamics of platelets activation and aggregation [24]. Although mathematical modelling provides an important tool for the qualitative understanding of the underlying mechanisms behind thrombus growth, it is difficult to properly use it to quantify the effects of these mechanisms.

Among the challenges that face the use of mathematical modelling as a tool to conduct robust quantitative studies of thrombus are the disparities in the values of blood coagulation kinetic constants reported in the literature. These disparities stem their origin from different settings in which these parameters are measured, which calls into question the validity of the predictions made by the models of thrombus formation [25]. Furthermore, difficulties in performing *in vivo* and *in vitro* experiments of thrombus growth represent an important challenge, that must be overcome for the validation of computational models. Moreover, most of the spatial models of thrombus formation are too complex and computationally expensive which limits their usefulness in conducting quantitative studies.

This work is devoted to data-driven modelling of thrombus formation under venous flow. Mathematical models of blood coagulation are formulated and used in order to investigate the effects of platelets on blood coagulability and on the initiation of thrombus formation under various conditions. We sketch numerical basics of the models since their computational aspects are beyond the scope of the present paper. The adopted approach consists in reproducing the available experimental data by calibrating the developed models and conducting numerical simulations. The calibrated models are simplified and analyzed to predict the clotting response under different conditions. This method is of particular importance because it can be used to perform quantitative studies for a wide range of model parameters without running long consecutive simulations using more sophisticated models. To evaluate the effect of platelet count on blood coagulability, we propose a new model, describing the kinetics of thrombin generation. The kinetic constants of the model were fitted to reproduce the observed thrombin generation in assay kits. We measure the values of the thrombin peak and the endogenous thrombin potential (ETP) for different densities of platelets and confirm the existing

relationship between high platelet count and hypercoagulability. The next part of the paper is devoted to the study of the effects of platelets on thrombus growth under venous flow. To do this, we calibrate a previously developed model of clot growth under flow [26, 27] to realistically simulate the *in vitro* experiments [11] in 3D settings. Among the main features of the considered model is the possibility to reduce it to a one-equation model yielding approximately the same solution. This offers the opportunity to conduct quantitative studies and easily evaluate the conditions of the initiation of clotting on various parameters. In particular, we estimate the thresholds of the shear rate that prevent clotting for different platelet count values as well as the critical size of the injured area leading to the initiation of clot growth.

## 2 Mathematical modelling of venous thrombus growth under flow

### 2.1 Spatiotemporal modelling of thrombus development

#### 2.1.1 Advection-diffusion-reaction equations for blood factors and platelet densities.
We extend the previously developed model of clot growth dynamics [26, 27] to include the aggregation of platelets. The model consists of advection-diffusion-reaction equations describing the spatiotemporal distributions of blood coagulation factors and platelet subtypes during thrombus development. A similar model was previously used to describe venous thrombus formation in rheumatoid and psoriatic arthritis [28] as well as during anti-vitamin K therapy [29]. The part of the coagulation cascade considered in the model is shown in Fig 1. First, we describe the concentration of prothrombin:

$$\frac{\partial P}{\partial t} + \nabla \cdot (\mathbf{u}P - D\nabla P) = -(k_1\phi_c + k_2 B_a + k_3 T + k_4 T^2 + k_5 T^3)P, \tag{1}$$

here and in what follows, $\mathbf{u}$ is the flow velocity and $D$ is the diffusion coefficient taken the same for all the clotting factors. The term in the right-hand side of the equation represents the activation of prothrombin by the active factors present on the surface of active platelets, factors IXa and Xa, and factors of the propagation phase respectively. Thrombin activates clotting factors of the propagation phase such as FV, FXIII, and FXI. Thus, the concentration of their active counterparts is proportional to thrombin concentration. The mathematical derivation of the polynomial terms describing the self-amplifying production of thrombin and their exact expressions are presented in the Appendix A. The rate of prothrombin consumption is equal to the rate of thrombin generation. Hence, we describe the concentration of thrombin as follows:

$$\frac{\partial T}{\partial t} + \nabla \cdot (\mathbf{u}T - D\nabla T) = (k_1\phi_c + k_2 B_a + k_3 T + k_4 T^2 + k_5 T^3)P - k_6 AT, \tag{2}$$

where the last term $-k_6 AT$ represents the inhibition of thrombin by prothrombin. Next, we describe the concentration of the clotting factors FIXa and FXa involved in the initiation phase. We denote their concentrations by $B_a$:

$$\frac{\partial B_a}{\partial t} + \nabla \cdot (\mathbf{u}B_a - D\nabla B_a) = k_7\phi_c(B^0 - B_a) + k_8 T(B^0 - B_a) - k_9 A B_a, \tag{3}$$

where the term in the right-hand side of the equation represents the activation of FIX and FX, $B = B^0 - B_a$ by active platelets. The second term describes their activation by thrombin and the last term accounts for their inhibition by antithrombin. We will present the boundary conditions for $B_a$ below. Now, let us describe the concentration of antithrombin:

$$\frac{\partial A}{\partial t} + \nabla \cdot (\mathbf{u}A - D\nabla A) = -k_6 AT - k_9 A B_a, \tag{4}$$

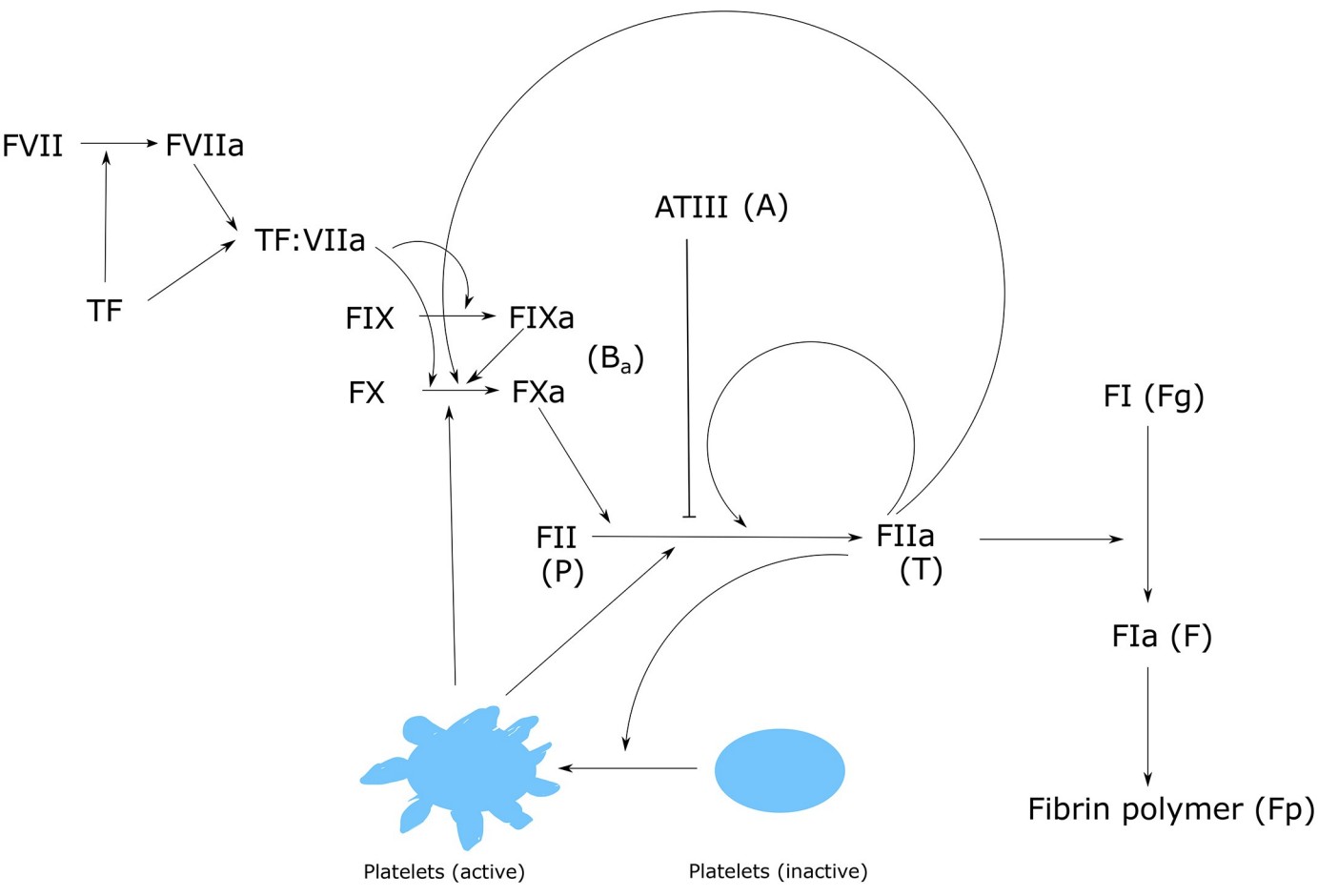

**Fig 1. Schematic representation of the part of the coagulation cascade described in the model.**

where the first and second terms in the right-hand side of the equation represent the anti-thrombin consumption during its inhibition of thrombin as well as factors IXa and Xa. Next, we describe the concentration of fibrinogen $F_g$:

$$\frac{\partial F_g}{\partial t} + \nabla \cdot (\mathbf{u}F_g - D\nabla F_g) = -\frac{k_{10}TF_g}{K_{10} + F_g}, \tag{5}$$

where the term $-\frac{k_{10}TF_g}{K_{10}+F_g}$ represents the rate of fibrinogen conversion to fibrin by thrombin. This rate of fibrinogen consumption is equal to the production rate of fibrin. Hence, we describe the concentration of fibrin as follows:

$$\frac{\partial F}{\partial t} + \nabla \cdot (\mathbf{u}F - D\nabla F) = \frac{k_{10}TF_g}{K_{10} + F_g} - k_{11}F, \tag{6}$$

where the last term in the right-hand side of the equation represents the conversion of fibrin into fibrin polymer. Fibrin polymer concentration is described as follows:

$$\frac{\partial F_p}{\partial t} = k_{11}F. \tag{7}$$

Fibrin polymer forms a solid clot. Hence, it does not diffuse and it is not transported by flow. Next, we describe the density of platelets in flow $\phi_f$ and in the clot $\phi_c$:

$$\frac{\partial \phi_f}{\partial t} + \nabla \cdot k(\phi_c + \phi_f)(\mathbf{u}\phi_f - D_p \nabla \phi_f) = -k_{12} T \phi_f - k_{13} \phi_f \phi_c,$$ (8)

$$\frac{\partial \phi_c}{\partial t} + \nabla \cdot k(\phi_c + \phi_f)(\mathbf{u}\phi_c - D_p \nabla \phi_c) = k_{12} T \phi_f + k_{13} \phi_f \phi_c,$$ (9)

where $k(\phi_c + \phi_f)D_p$ is the effective diffusion coefficient for platelets, $k(\phi_c + \phi_f)$ is a decreasing function given by $k(\phi_c + \phi_f) = \tanh\left(\pi\left(1 - \frac{\phi_c + \phi_f}{\phi_{max}}\right)\right)$. The behaviour of platelets in the flow depends on concentration of platelets [22] and resembles that of traffic flow. We take into account that thrombin activates platelets in the flow and contributes to their attachment to the clot. These platelets can also become active if they get into contact with the active platelets in the clot.

**2.1.2 Boundary conditions.**   We apply the zero-flux condition at the all boundaries for thrombin, fibrin, and fibrin polymer. The concentrations of prothrombin, antithrombin, and fibrinogen were assumed to be at the physiological level in the plasma at the inlet. We set the concentration of platelets in the flow to $\phi_f = \phi_f^0$ as the initial condition and at the influx boundary. We apply the condition $\phi_c = \phi_c^0$ at the injured area to represent the initial concentration of active platelets bound to the subendothelium and the zero-flux boundary condition at the rest of the boundary.

In order to describe the generation of thrombin in the initiation phase at the damaged endothelial wall, we consider the complex $T_F^*$ formed by the tissue factor and factor VII. Factors IX and X interact with this complex due to a surface reaction. They come from the bulk solution being inactive, form a complex $[T_F^*B]$ with $T_F^*$ and return to the solution in the active form. The reaction rate for the surface concentration of this complex can be written as follows:

$$D\frac{\partial [T_F^*B]}{\partial \mathbf{n}} = k_f^+ B\left(T_F^* - [T_F^*B]\right) - k_f^- [T_F^*B],$$

where the first term in the right-hand side of this equation describes the flux of $B$ to the surface, the second term describes the flux from the surface. Assuming that this reaction is fast, we can use the detailed equilibrium:

$$k_f^+ B(T_F^* - [T_F^*B]) = k_f^- [T_F^*B].$$

Then

$$D[T_F^*B] = \frac{k_f B T_F^*}{1 + k_f B},$$ (10)

where $k_f = k_f^+ / k_f^-$.

The boundary conditions for the variables $B$ and $B_a$ at the damaged surface are as follows:

$$D\frac{\partial B}{\partial \mathbf{n}} = -k_f^+ B(T_F^* - [T_F^*B]), \quad D\frac{\partial B_a}{\partial \mathbf{n}} = k_f^- [T_F^*B].$$ (11)

We prescribe the zero flux condition at the intact surface.

Assuming that the sum of the concentrations of activated and inactivated factors remains approximately constant because the rate of factors IXa and Xa consumption by antithrombin

is very low, $B_a + B = B^0$, we obtain from (10) and the second equality in (11):

$$\frac{\partial B_a}{\partial \mathbf{n}}\Big|_{\Gamma_d} = \frac{\alpha_1(B^0 - B_a)}{D(1 + \beta_1(B^0 - B_a))}, \tag{12}$$

where $\alpha_1 = k_f^- k_f T_F^*$ and $\beta_1 = k_f$.

In the case of nonzero flow velocity, the concentrations of prothrombin, fibrinogen and antithrombin at the entrance of the domain $\mathbf{x} = 0$ are kept constant.

**2.1.3 Blood flow model.**   We consider blood plasma as an incompressible Newtonian fluid and use the Navier-Stokes equations to describe the dynamics of blood flow in the vessel with rigid walls and its influence on clot growth dynamics:

$$\rho \frac{\partial \mathbf{u}}{\partial t} + \text{div} \left( \rho \mathbf{u} \mathbf{u}^T - \mu \nabla \mathbf{u} + Ip \right) = -\frac{\mu}{K_f} \mathbf{u},$$

$$\text{div} \left( \mathbf{u} \right) = 0, \tag{13}$$

where $\mathbf{u}$ is the flow velocity contributing to (1)–(9), $p$ is the pressure, $\rho$ is the density of the blood, $\mu$ is the dynamic viscosity, $K_f$ is the hydraulic permeability of the clot [6]:

$$\frac{1}{K_f} = \frac{16}{\alpha^2} \widetilde{F}_p^{\frac{3}{2}} \left( 1 + 56\widetilde{F}_p^3 \right) \left( \frac{\phi_{max} + \phi_c}{\phi_{max} - \phi_c} \right), \tag{14}$$

here $\widetilde{F}_p = \min\left(\frac{7}{10}, \frac{F_p}{7000}\right)$ is the normalized concentration of fibrin polymer in the clot, $\alpha$ is the radius of the fibers.

Blood flow is driven by the pressure difference, we prescribe the pressure $p_{in}$ at the inlet $\Gamma_{in}$ and the pressure $p_{out}$ at the outlet $\Gamma_{out}$ and no-slip boundary condition $\mathbf{u} = \mathbf{0}$ at the rest of the boundary $\partial\Omega\backslash(\Gamma_{in} \cup \Gamma_{out})$. To set the inlet pressure in dependence on shear rate parameter $\gamma$, we use the formula $p_{in} = 4L\gamma\mu/D$, where $L$ is the length of the vessel and $D$ is the diameter of the vessel. The outflow pressure is set to zero $p_{out} = 0$. Note that red blood cells (RBCs) and their effects are not considered in the present work because the microfluidic experiments that are used to validate the model are performed on blood plasma which does not contain these cells. Contrary to the condition of fixed inlet velocity, the imposed pressure condition allows the clot to completely occlude the vessel and obstruct the flow.

**2.1.4 Numerical implementation of the model.**   The computational grid for a vessel of 8 *mm* length and 1 *mm* diameter is shown in Fig 2. The TF patch, with the width 0.2 *mm*, is located at 2 *mm* from the vessel inlet. The grid consists of 20160 hexahedral cells and it is refined towards the surface of the cylinder (Fig 2B). The refinement is required for more accurate recovery of the distribution of platelets in the vicinity of the vessel wall and the fibrin polymer formation.

We use a novel cell-centered finite volume solver with a collocated arrangement of the velocity, pressure and all the clotting factors. The discrete problem is solved in a fully-coupled fully-implicit manner using finite volume discretizations.

The Eq (13) may be rewritten in the form $\text{div}(\mathbf{A}) = \mathbf{g}$ where $\mathbf{A}$ is the flux depending on $\mathbf{u}$ and $p$. Integrating over each cell $\omega$ of the computational mesh and applying Green-Gauss theorem we get

$$\int_\omega \text{div}(\mathbf{A})\text{d}\omega = \oint_{\partial\omega} \mathbf{A} \cdot \mathbf{n}\text{d}S = \int_\omega \mathbf{g}\text{d}\omega$$

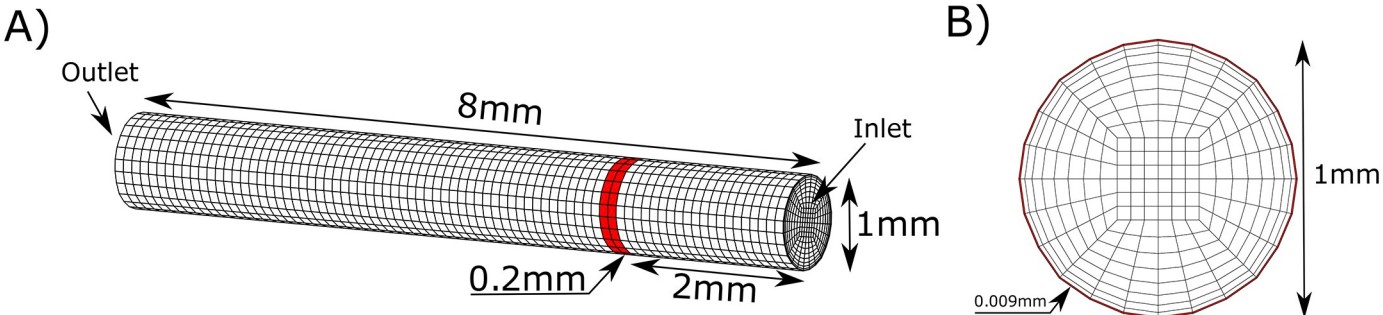

**Fig 2.** A) The settings for 3D numerical simulations of the coagulation process. Red strip at the surface indicates the TF patch with the width of 0.2 mm. B) Cross-section of the computational domain. The mesh is refined towards the vessel wall. The width of the numerical cells at the boundary is approximately 0.009 *mm*.

which is reformulated in terms of the vector flux $\mathbf{t} = \mathbf{A} \cdot \mathbf{n}$

$$\sum_{\text{faces } f \text{ of cell } \omega} \mathbf{t}|f| = \mathbf{g}|\omega|.$$

(15)

Eq (13) and vector representation of unknowns $(\mathbf{u}, p)^T$ yield decomposition of $\mathbf{t}$

$$\mathbf{t} = \begin{pmatrix} \rho \mathbf{u}\mathbf{u}^T\mathbf{n} - \mu\nabla\mathbf{u}\mathbf{n} + p\mathbf{n} \\ \mathbf{n}^T\mathbf{u} \end{pmatrix} = \begin{pmatrix} \rho \mathbf{u}\mathbf{u}^T\mathbf{n} \\ 0 \end{pmatrix} - \begin{pmatrix} \mu\nabla\mathbf{u}\mathbf{n} \\ 0 \end{pmatrix} + \begin{pmatrix} 0 & \mathbf{n} \\ \mathbf{n}^T & 0 \end{pmatrix} \begin{pmatrix} \mathbf{u} \\ p \end{pmatrix}.$$

(16)

Plugging discretizations of each term in (16) into the cell balance Eq (15) defines the finite volume scheme. The details of the discretization are given elsewhere [30]. The method is stable, second-order accurate and has a compact computational stencil composed of cells neighbouring through face. The accuracy of the scheme was verified by comparisons with the analytical solutions of the Ethier-Steinman [31] and Poiseuille flows. Advection-diffusion-reaction Eqs (1)–(9) are discretized by a positivity preserving cell-centered finite volume scheme [32, 33] which in our case has the first-order accuracy due to the single point upstream approximation of the advective flux. The finite volume scheme for platelets Eqs (8) and (9) use more elaborated stable flux discretization. The simulation was performed on INM RAS cluster. The tools for automatic differentiation, parallel mesh management, parallel linear and nonlinear solvers are provided by the INMOST library [34]. More details of the numerical schemes discretizing Eqs (1)–(9) and the simulation are given in [30], computational issues are beyond the scope of the present paper.

## 2.2 Thrombin generation model

Thrombin generation tests are widely used for the evaluation of blood coagulation in individual plasma samples. These tests measure the concentration of generated thrombin over time after the introduction of an amount of TF to plasma. We suggest a new model for thrombin generation in the presence of platelets in the motionless plasma. We first describe the concentration of FXa, $[Xa]$ using the following equation:

$$\frac{d[Xa]}{dt} = a_1 TF[VIIa]\big([X]_0 - [Xa]\big) + a_2[IIa]\big([X]_0 - [Xa]\big)$$

$$+ a_3 \phi_0 [IIa]([X]_0 - [Xa]) - a_4[Xa][ATIII],$$

(17)

where the *TF* and $[VIIa]$ denote the concentrations of tissue factor and FVIIa respectively. The first term in the right-hand side of this equation represents the activation of FX by the complex

TF-FVIIa. The second term describes the activation of FX by thrombin. The third term gives the activation rate of FX by platelets whose activation, in its turn, depends on thrombin. The last term describes the inhibition of FXa by antithrombin [$ATIII$].

Next, we describe the concentration of prothrombin [$II$]:

$$\frac{d[II]}{dt} = -\left(b_1[Xa] + k_3[IIa] + k_4[IIa]^2 + k_5[IIa]^3\right)[II], \tag{18}$$

where the right-hand side describes the consumption of prothrombin upon its conversion to thrombin. Prothrombin is activated by FXa in the initiation phase and by other factors, such as FV, FVIII, and FXI in the propagation phase. These factors are activated by thrombin [$II$]. Therefore, the density of their active form is proportional to thrombin concentration. We describe the concentration of thrombin [$IIa$] by the following equation:

$$\frac{d[IIa]}{dt} = \left(b_1[Xa] + k_3[IIa] + k_4[IIa]^2 + k_5[IIa]^3\right)[II] - b_2[ATIII][IIa]. \tag{19}$$

The rate of thrombin production is equal to the consumption rate of prothrombin. The second term in the right-hand side of this equation represents the direct inhibition of thrombin by antithrombin. The concentration of the latter [$ATIII$] is described as follows:

$$\frac{d[ATIII]}{dt} = -a_4[Xa][ATIII] - b_2[IIa][ATIII]. \tag{20}$$

Initial concentrations of blood factors and the values of reaction rate constants are given in Table 1 in the Appendix B. The reaction rates in the model (17)–(20) (without platelets) are chosen by fitting thrombin generation curves obtained in a more complete model [14] with known parameters. Then, the parameter $a_3$ was fitted to reproduced the experimental results.

## 2.3 A simplified model of thrombin distribution

Thrombin plays an important role in the coagulation process as it converts fibrinogen into fibrin and then to fibrin polymer forming the clot. Similarly to our previous works [26, 27], we derive the following simplified 1D model for thrombin distribution from the spatiotemporal model of thrombus formation presented in Section 2.1:

$$\frac{\partial T}{\partial t} = D\frac{\partial^2 T}{\partial y^2} + \Phi(T, y), \tag{21}$$

where

$$\Phi(T, y) = (k_1\phi_0 + k_2 B_a(y) + k_3 T + k_4 T^2 + k_5 T^3)(P_0 - T) - \sigma(y)T,$$

$D$ is the diffusion coefficient, $P_0$ is the prothrombin concentration initially present in blood flow, $k_1$, $k_2$, $k_3$, $k_4$, and $k_5$ are kinetic coefficients. The function $\sigma(y) = k_6 A_0 + a\gamma$ shows the influence of antithrombin $A_0$ and of the blood flow in the downregulation of thrombin production. Here $\gamma$ is the shear rate. Various simplifying hypotheses were considered in the derivation of the model. First, we suppose that consumption of clotting factors such as prothrombin and antithrombin does not significantly affect thrombus formation. Next, we assume that clot growth does not affect the local hemodynamics, and that the latter exerts a constant shear rate during the whole process of blood coagulation. Furthermore, we suppose that $B_a(y)$ is the stationary solution of Eq (3). To simplify the interpretation of the model, we do not consider the activation of $B_a(y)$ by $T$ and $\phi_0$ at this stage. The general solution is then given by the form: $B(y) = B^0 e^{-\sigma y}$. The production of thrombin by platelets and its amplification are directly considered in the

thrombin equation. Note that the simplified 1D model (21) exhibits the same dynamics and the same speed of thrombin propagation as the more complete model [27–29]. Hence, the model approximates the evolution of the maximal height reached by the distribution of thrombin over time. Therefore, it provides a tool to describe the rate of vessel occlusion.

## 2.4 Identification of the parameters

We use a data-driven approach to calibrate the parameters of the models. First, we consider coagulation kinetics in the absence of platelets. The reaction rate constants for this system were taken from the literature except for the parameters $k_3$, $k_4$, and $k_5$ which are specific to the model. These contants can be written as a function of the activation and inhibition rates of the reactions that participate in the amplification of thrombin production (Appendix A). They were identified using the thrombin generation model by comparing with data in reference [14] using a Monte-Carlo algorithm to speed up the process. Then, we fitted the values of physiological parameters (blood clotting concentrations) to reproduce the experimental results describing the initiation time for NPP [11]. The rate of thrombin activation by platelets was fitted by comparing the output of the simplified model with the experimental data for the PRP [11], while the reaction constants for platelet dynamics $k_{12}$ and $k_{13}$ were taken from the literature [22, 35]. The values of the parameters used in the model of thrombus formation in flow are provided in Table 2 in the Appendix B.

## 3 Numerical results

### 3.1 Quantification of the relationship between platelet count and blood coagulability

In order to evaluate the relationship between platelet count and blood coagulability, we use the thrombin generation model from Section 2.2 to reproduce the experiment conducted in reference [39]. These experiments consist of monitoring the generation of thrombin in mice blood with different platelet count after the introduction of 1 $pM$ of TF. Thrombin generation curves can be characterized by different values such as the peak value and the endogenous thrombin potential (ETP) which corresponds to the total value of generated thrombin over time. The study shows the existence of a linear relationship between platelet count and the measured peak thrombin value. A hyperbolic relationship was also observed between platelet count and the ETP value. The model of thrombin generation was able to successfully reproduce these results as shown in Fig 3. Indeed, the similarity between the simulated and observed peak value and ETP implies that the thrombin generation curves are in a good agreement as well. We speculate that the small differences between the measured values and the predicted ones can be due to the intersubject variability in such experiments. In blood coagulation, the propagation phase is triggered when the generated value of thrombin reaches a certain threshold. The model predicts that the peak concentration of generated thrombin reaches 141.4 $nM$ when the platelet count corresponds to $350 \times 10^9 L^{-1}$. This value is sufficient to provoke thrombin wave propagation according to the simplified model (Eq (21)).

### 3.2 The shear rate threshold that prevents clotting initiation depends on the density of platelets

To investigate the dynamics that characterizes the initiation of clotting in blood flow, we use the model of thrombus growth (1)–(14) to reproduce the experiment on *in vitro* clot growth [11]. In this experiment, blood plasma can be described by a Newtonian flow driven by a constant pressure drop. The computational domain corresponds to the tube with rigid walls.

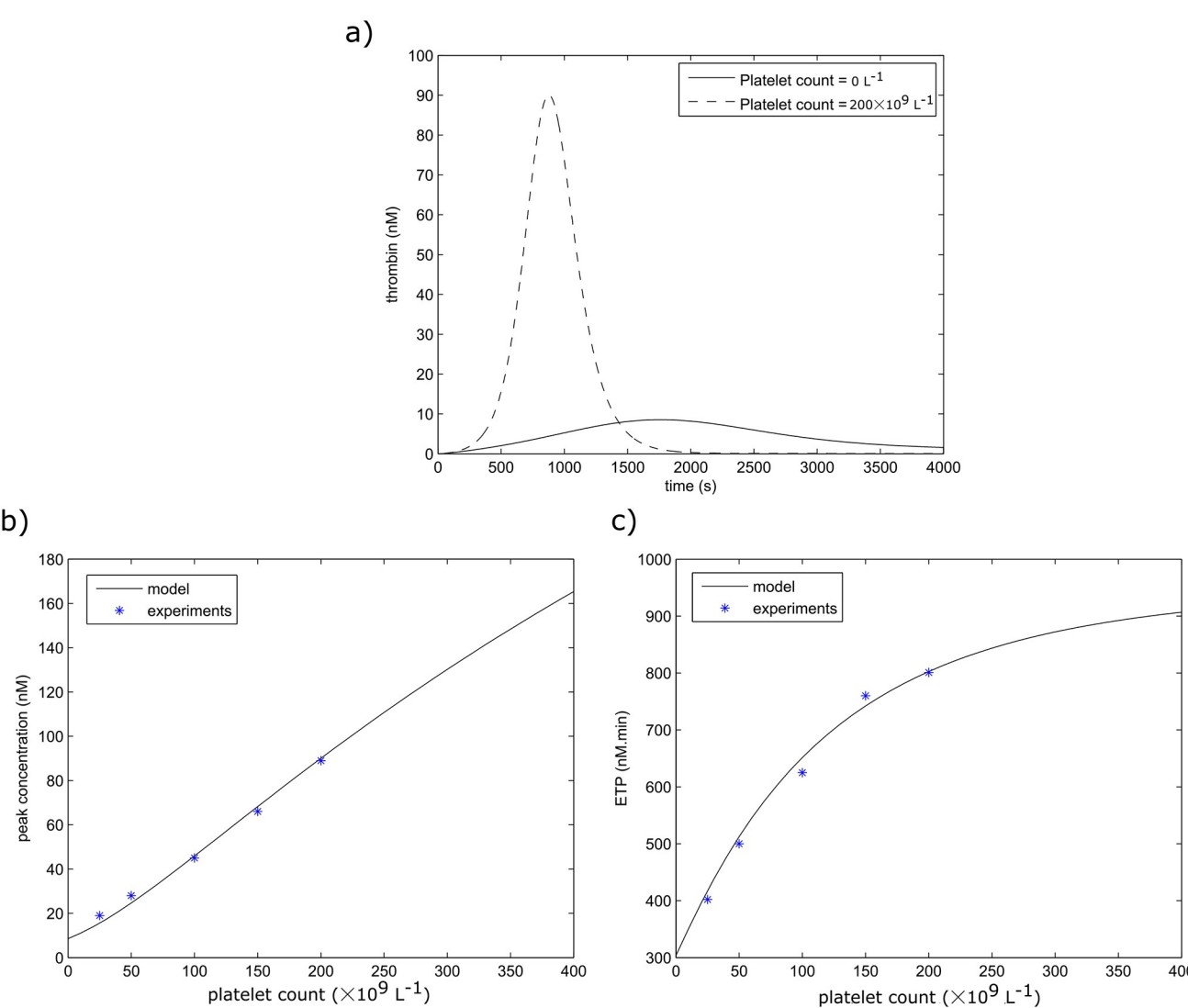

**Fig 3.** a) Thrombin generation curves for two values of platelet count. b) The relationship between thrombin peak value and platelet count according to model predictions and experiments. c) The relationship between ETP and platelet count as reported by the model and experiments.

We run numerical simulations for a physical time corresponding to 1200 *s* and determine the initiation time of blood coagulation corresponding to the beginning of the propagation phase when a sudden elevation of the maximal value of generated fibrin polymer near the TF patch can be observed. This moment coincides with the time when the vessel is no longer completely permeable and the clot will subsequently occlude the vessel. We vary the value of the pressure drop to control the initial shear rate. Then, the value of the shear rate can also change if clot growth is initiated. However, in this case we intend to determine the value of the shear rate that prevents the initiation of the coagulation process. Therefore, we determine the value of the initial shear rate that prevents blood clotting by running consecutive simulations. We consider that for PRP, platelet density is equal to $300 \times 10^9 \ L^{-1}$ which is the value considered in the experiments [11]. Similarly to the findings of the experiments, a threshold response of clotting initiation to the shear rate was observed in the numerical simulations. When the shear rate is lower than a threshold value, coagulation is initiated at an early time before $t = 800 \ s$

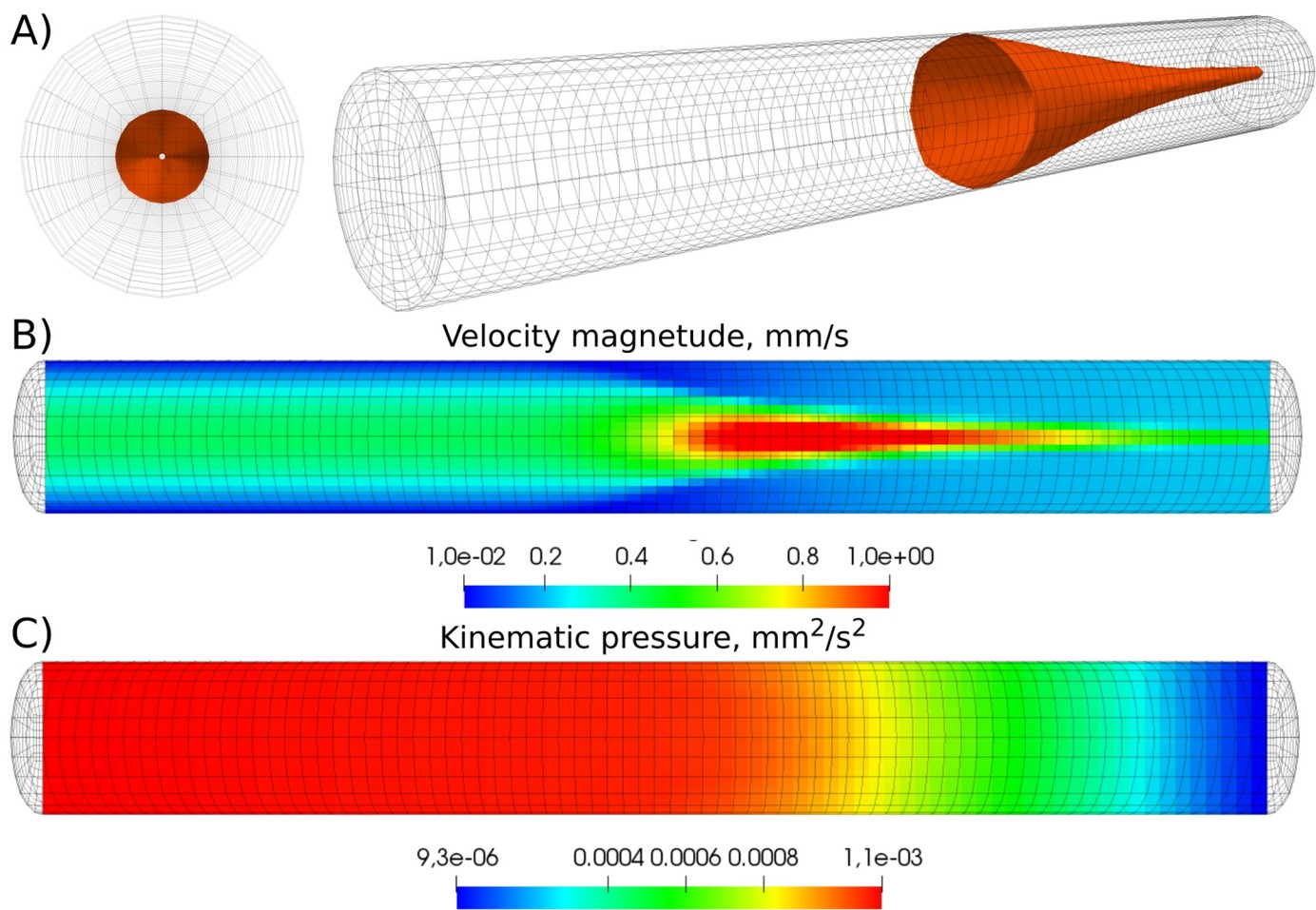

**Fig 4. Fluid dynamics before vessel occlusion for platelet-rich plasma (PRP) for the shear rate $\gamma = 25 \ s^{-1}$ and at the time $t = 60 \ s$.** The vessel is still permeable, although the flow is significantly obstructed by the clot. A) isosurface of fibrin polymer corresponding to the hydraulic resistance $1/K_f = 10 \ mm^{-2}$. B) The magnitude of velocity in the log-scale. C) The distribution of pressure.

and the thrombus occludes the vessel. In this case, blood flow can be completely stopped and the stagnant plasma turns into a thrombus. When the value of the shear rate is sufficiently high, the initiation of the coagulation process takes a longer time (higher than $t = 800 \ s$). Figs 4 and 5 represent the results of numerical simulations of thrombus formation for platelet-rich plasma at the shear rate $\gamma = 25 \ s^{-1}$ and times $t = 60 \ s$ and $t = 70 \ s$, respectively. For illustration purposes, the velocity magnitude is demonstrated in the log-scale. From the velocity distribution, it is evident that the flow is obstructed by the clot at time $t = 70 \ s$. Both figures show that the model is symmetrical with respect to the azimuth angle, but not with regard to the height of the cylinder.

In order to compare the simulation results with the experiments in the *in vitro* microfluidic chambers [11], we compute the necessary time for blood clotting initiation while increasing the shear rate $\gamma$. A threshold response is observed in the initiation of blood clotting characterized by a critical value which is higher in platelets rich plasma (Fig 6A). We observed that the value of the clotting initiation time becomes stable for high shear rates in long tubes. However, it keeps increasing exponentially for short tubes. The simplified model predicts a sharp transition between the regimes of complete and partial occlusion when the shear rate exceeds the

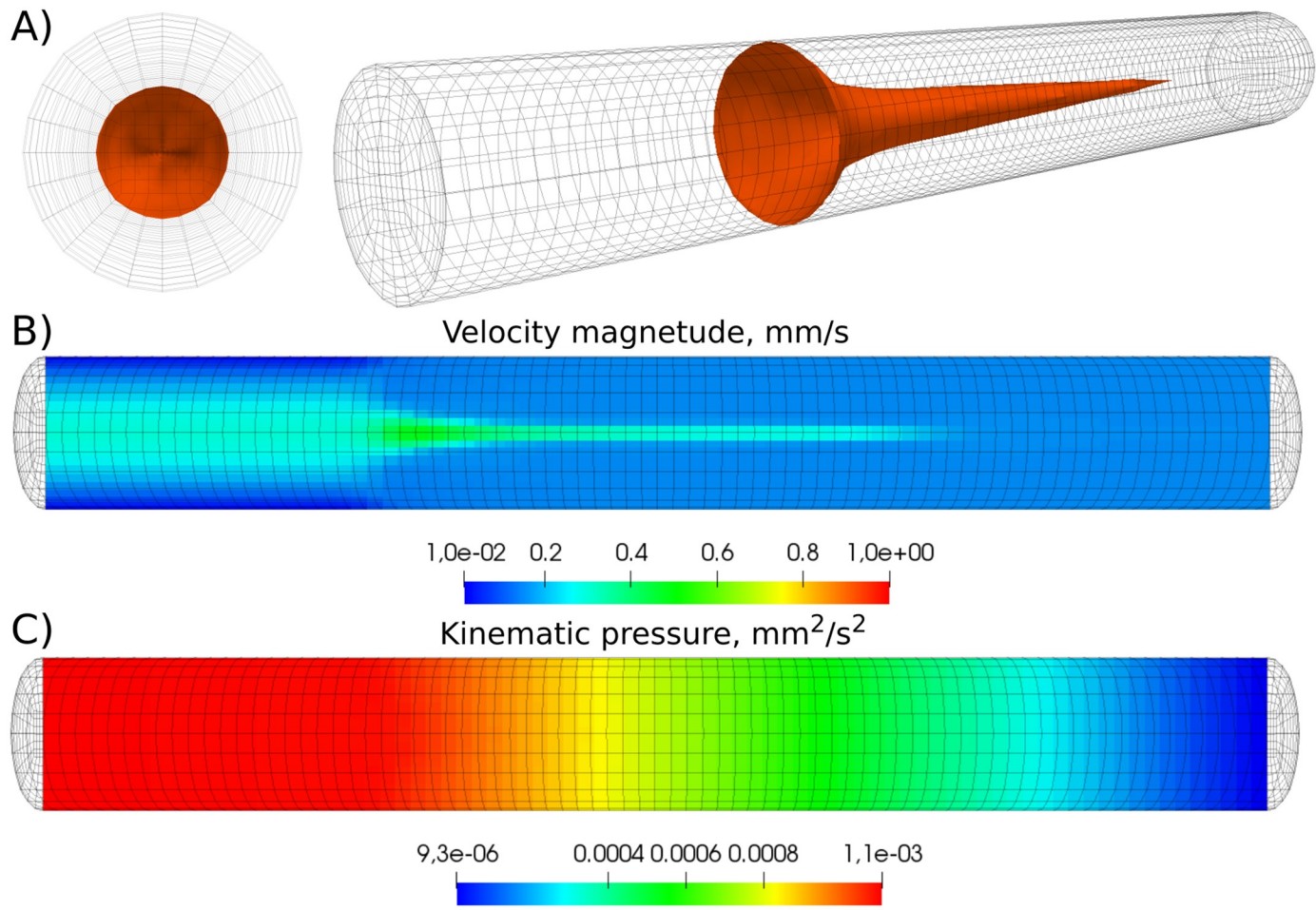

**Fig 5. Fluid dynamics after the occlusion of the vessel for platelet rich-plasma for the shear rate $\gamma = 25 \, s^{-1}$ and at the time $t = 70 \, s$.** The vessel is already impermeable due to the clot. A) isosurface of fibrin polymer corresponding to the hydraulic resistance $1/K_f = 10 \, mm^{-2}$. B) The magnitude of velocity in the log-scale. C) The distribution of pressure.

threshold value (Fig 6B). These critical values are in good agreement with the ones reported in the experiments (Fig 6A) and with the results of the full 3D model (Fig 6C).

While numerical simulations can be used to have a realistic representation of the process of clot growth, the simplified model for thrombin distribution (21) provides a powerful tool to analyze the various mechanisms involved in the development of venous thrombi. The production of thrombin is followed by the formation of the fibrin mesh. Therefore, we consider that the percent of the vessel that is occluded corresponds to the regions where thrombin exceeds a certain threshold $T^* = 200 \, nM$. For Eq (21), it is known that the wave speed is positive if and only if

$$\int_{T_0}^{T_2} \Phi(T, y) dT > 0. \tag{22}$$

Here $T_0$ and $T_2$ are stable zeros of the function $\Phi(T, y)$. Note that this condition is verified for a fixed $y$ assuming that the reaction front is narrow. This condition also gives the clot size which corresponds to the value of $y$ where the clot stops growing. The condition on shear rate for

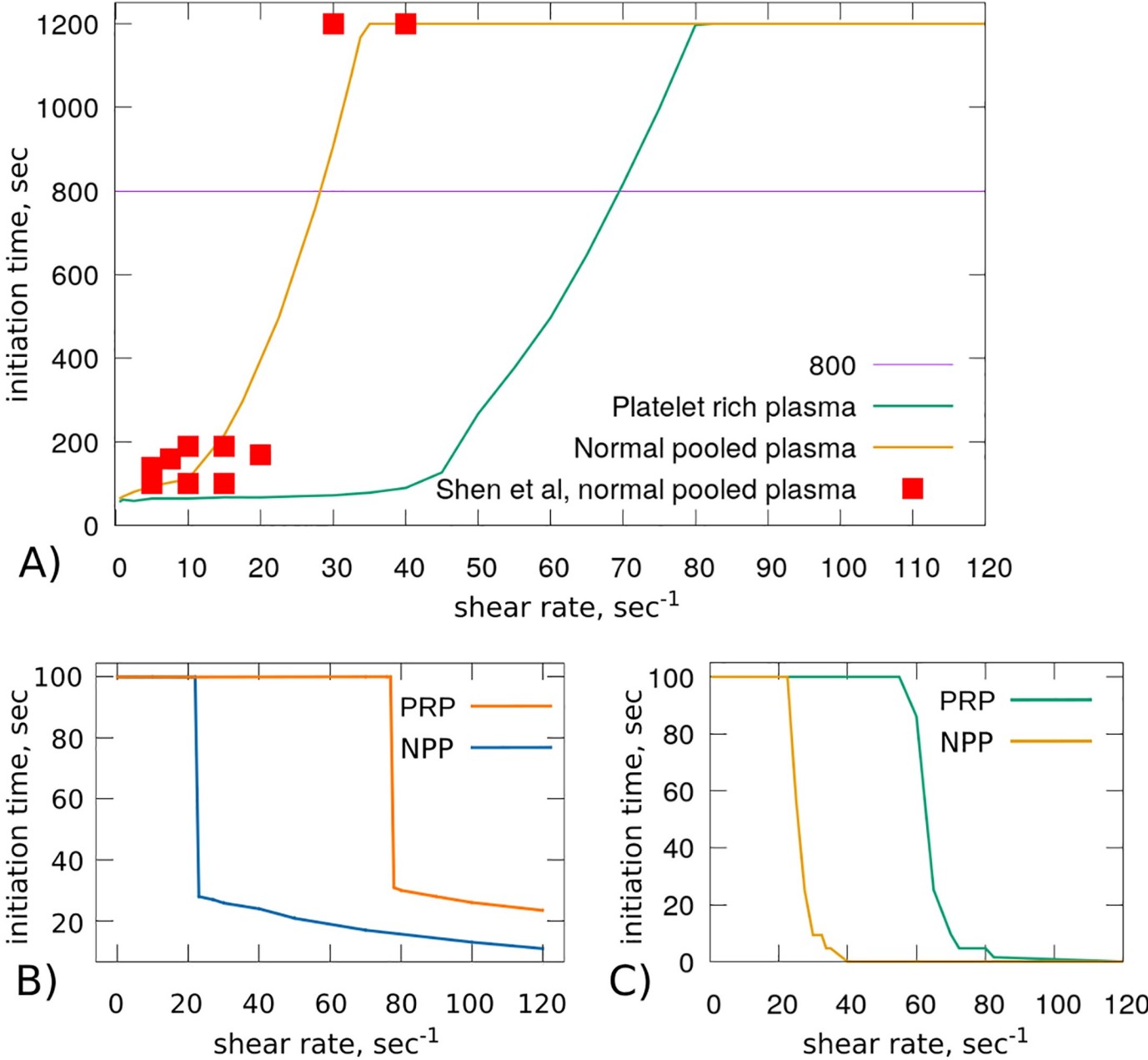

**Fig 6.** A) The moment of clotting initiation as a function of the initial wall shear rate for NPP and PRP for a long tube with the length 8 *mm*. Experimental results of *in vitro* thrombus formation are shown for comparison purposes (reproduced from [11, 40]). B) The percentage of the vessel occupied by the clot at $t = 800\,s$ according to the one-equation 1D model. Occluded regions are domains where teh concentration of thrombin exceeds 200 *nM*. C) The percentage of the occluded vessel at $t = 800\,s$ according to the full 3D model.

thrombin propagation is as follows:

$$\gamma > \gamma^*, \tag{23}$$

where

$$\gamma^* = \frac{1}{aT_2/2}\left(\begin{array}{l} k_1(\phi_0 P_0 T_2/2 - \phi_0 T_2^2/3) + k_2(B_a(y)P_0 - B_a(y)T_2/2) + k_3(P_0 T_2^1/2 - T_2^2/3) + \\ k_4(P_0 T_2^4/3 - T_2^3/4) + k_5(P_0 T_2^3/4 - T_2^4/5) - k_6 A_0 T_2/2 - av(y)T_2/2 \end{array}\right).$$

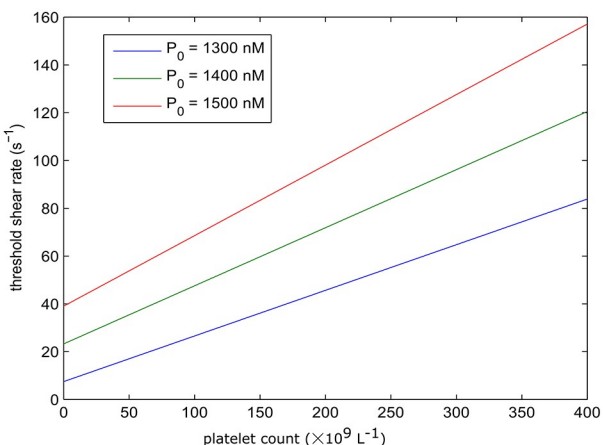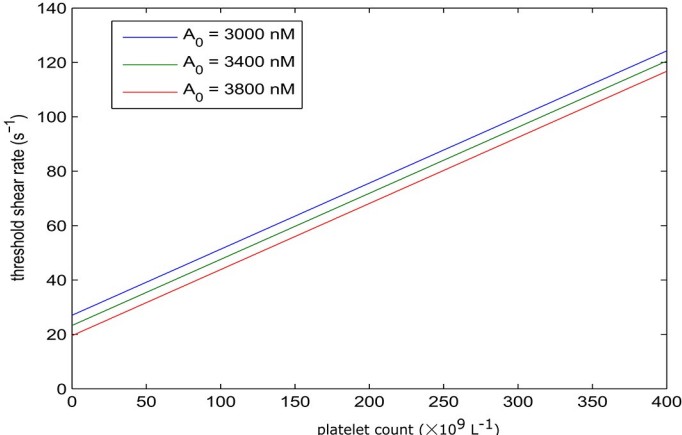

**Fig 7. The shear rate threshold of clotting initiation as a function of platelets density for different prothrombin (left) and antithrombin concentrations (right).**

The threshold of shear rate that initiates the coagulation process, is linearly proportional to the platelet density for various values of prothrombin and antithrombin concentrations as shown in Fig 7. The concentration of prothrombin has a greater impact on the this linear relationship than the concentration of antithrombin.

### 3.3 Quantification of the effect of TF surface area on the initiation of blood clotting

The surface of exposed TF is another important factor that determines the necessary time for blood clotting initiation. To explore the effects of this parameter on the coagulation process, we conduct systematic numerical simulations for different sizes of the TF patch with NPP. We impose a fixed shear rate equal to $\gamma = 40\ s^{-1}$ in all these simulations. Increasing the size of the TF patch increases the production of FIXa and FXa which upregulates the generation of thrombin and accelerates the clotting initiation. As expected, numerical simulations show the existence of a threshold value for the TF patch that initiates blood coagulation (Fig 8). As before, the value above which the initiation of blood clotting can be observed is very close to the one observed in experiments. This threshold response to the size of the TF patch was also observed in other experimental studies [41, 42].

## 4 Discussion

This work presents a quantitative study of the effects of platelet count, wall shear rate, and injury size on the initiation of blood coagulation based on the analysis of the model and on numerical simulations. The calibration of previously developed models allowed us to describe the results of clinical studies and *in vitro* experiments in microfluidic chambers. To evaluate the effects of platelets on blood coagulability, we have simulated the thrombin generation curves for blood samples with variable platelet counts. Similarly to the experiments [39], a linear relation was observed between the platelet count and the peak value reached by thrombin concentration. Furthermore, the existence of a positive hyperbolic relation between the platelet count and the ETP was confirmed. These relationships are in good agreement with the clinical indications that consider a high platelet count as an independent risk factor for VTE while a

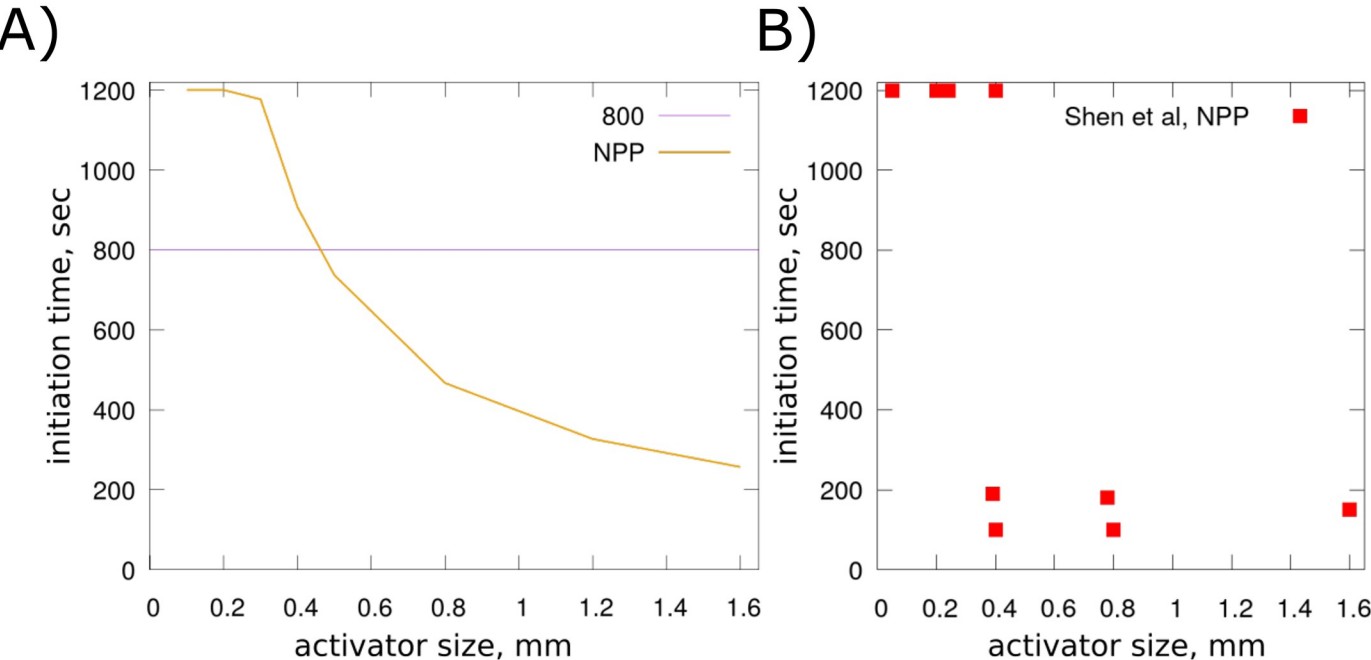

**Fig 8. A) The NPP and PRP clotting initiation time as a function of the TF patch size for a constant shear rate equal to 40 s$^{-1}$.** B) Experimental data showing the time of clotting initiation for different sizes of the activator [11].

low platelet count (thrombocytopenia) is considered to be associated with hemorrhagic events [43].

Hypercoagulability caused by high platelet count also accelerates the process of thrombus formation which increases the risk of venous thrombosis incidence. To explore the conditions that trigger venous thrombosis, we used a previously developed model of blood coagulation to reproduce a set of experiments of thrombus growth in microfluidic capillaries [11]. A novel 3D finite-volume solver was developed and used to mimic the settings of these experiments. The plasma was described as a Newtonian fluid driven by a constant non-pulsative pressure difference. Similarly to the experiments, two types of plasma were considered: normal-pooled (or platelet poor) plasma (NPP) and platelet-rich plasma (PRP). The most important finding of these experiments is confirming that the initiation of coagulation displays a threshold response to shear rate. This was also observed in our numerical simulations and was accurately reproduced by choosing the appropriate physiological parameters. As expected, the shear rate value that is necessary to prevent the initiation of thrombus growth was higher in PRP. We speculate that for a constant pressure drop, decreasing the length of the vessel or increasing its diameter result in an increase of the shear rate that prevents clotting initiation. Using a simplified model, we were able to specify this value for a wide range of platelet count values as well as for large sets of physiological parameters. A similar threshold response of the clotting initiation was also observed for the size of the TF patch. This phenomenon was already investigated in another computational modelling study [44]. Another important result obtained both in modelling and in the *in vitro* experiments was that the initiation of blood coagulation depends on shear rate and not on blood flow velocity (i.e. linearly related to blood flow velocity divided by the diameter of the blood vessel).

In the present study, we used data-driven modelling to quantify the effects of platelet count, shear rate, and other pathophysiological parameters on the pathogenesis of venous thrombosis. The synergy between the presented models allowed us to study the combined effects of various interlinked factors of venous thromboembolism such as blood coagulability, hemodynamics, and the size of the TF patch. These aspects represent the three elements of the Virchow's triad [10]. We have conducted experimentally validated numerical simulations of thrombin generation and thrombus development in the flow. At the same time, we have used the one-equation model of thrombin distribution to derive analytical estimates of the shear rate threshold which prevents clotting initiation for different plasma compositions. This model reduction replaced the parameter sensitivity analysis of the model of thrombus formation under the flow, which would require a much higher computational cost. Indeed, the main feature of the presented model in comparison with the rest of spatio-temporal models is the possibility to derive from it a thrombin generation model and a 1-D thrombin distribution model. As a result, it was possible to simulate the development of the thrombus under flow, to estimate the coagulability of blood, and to derive analytical estimates for the relationships between the initiation of coagulation and the various model parameters. Furthermore, there exist very few 3D continuous models that describe hemodynamics, the coagulation biochemistry, and platelet kinetics at the same time. The combination of the three models was also essential for the identification of parameters. For example, by fitting the thrombin generation model with the experimental curves, we could determine the parameters that describe the propagation phase in the two other models.

In order to reproduce experiments, some simplifying assumptions were imposed. They include the Newtonian nature of flow (blood plasma in this case), the non-pulsatile continuous aspect of flow, and the rigidity of the vessel wall (microfluidic capillaries). The similarity between the modelling conditions and these settings not only gave credibility to the quantitative predictions of the model but also allowed us to calibrate the model and to use it for the upcoming studies. The effects of RBCs can be taken into consideration implicitly through the choice of blood rheological properties. Their participation in the coagulation process consists in marginalizing platelets to the vessel wall. Thus, they can accelerate the clotting process [45], but this will not change the qualitative predictions made by the model and experiments that consider blood plasma. In fact, the threshold response to the TF patch size was observed under blood flow that contains RBCs [42] as well. The marginalization of platelets by RBCs can be introduced implicitly by using discrete methods such as dissipative particle dynamics [46] and immersed boundary methods [47]. Furthemore, blood can be simulated as a multi-constituent fluid with one phase for the plasma with RBCs and the other phase representing the plasma with platelets [48]. Another limitation of this study concerns the origin of blood in the experimental studies that were used to calibrate the model. Human blood was considered in the microfluidic experiments of clotting initiation [11], whereas mice plasma was used in the experiments of thrombin generation [39]. The mice plasma contains approximately the same concentrations of blood factors as its human counterpart [49]. The major differences between the two are the density of platelets and the concentrations of anticoagulant factors. In the experiments [39], the concentrations of antithrombin and TFPI were reduced by diluting the plasma and the density of platelets was varied in order to quantify its impact on thrombin generation. In the forthcoming works, we will quantify the effects of different physical and pathophysiological parameters on clot formation in non-Newtonian, pulsatile flow and inside elastic blood vessels. Furthermore, we will also use the model to reproduce other microfluidic experiments corresponding to other blood clotting scenarios'.

## Appendix A: Derivation of the model

Let us consider the following model which describes the self-amplifying production of thrombin:

$$\frac{\partial [Va]}{\partial t} = D\Delta [Va] + \hat{k}_1 T - h_1 [Va], \tag{24}$$

$$\frac{\partial [VIIIa]}{\partial t} = D\Delta [VIIIa] + \hat{k}_2 T - h_2 [VIIIa], \tag{25}$$

$$\frac{\partial [XIa]}{\partial t} = D\Delta [XIa] + \hat{k}_3 T - h_3 [XIa], \tag{26}$$

$$\frac{\partial [IXa]}{\partial t} = D\Delta [IXa] + \hat{k}_4 [XIa] - h_4 [IXa], \tag{27}$$

$$\frac{\partial [Xa]}{\partial t} = D\Delta [Xa] + \hat{k}_5 [IXa] + \hat{k}_{55} [VIIIa][IXa] - h_5 [Xa], \tag{28}$$

$$\frac{\partial T}{\partial t} = D\Delta T + \left( \hat{k}_6 [Xa] + \hat{k}_{66} [Xa][Va] \right) P - \sigma T, \tag{29}$$

derived from another model of blood coagulation in plasma [50]. The coefficients $\hat{k}_i$ denote the activation coefficient rates and $h_i$ the inhibition rate. The factors Va and Xa form the pro-thrombinase complex Va-Xa while the factors VIIIa and IXa for the complex VIIIa-IXa. They are considered in (28) and (29) in the form of the terms $\hat{k}_{55} [VIIIa][IXa]$ and $\hat{k}_{66} [Xa][Va]$, which are obtained using the assumption of detailed equilibrium for large reaction constants. The model can be completed by adding the equations for platelets, prothrombin, antithrombin, fibrinogen, fibrin, and fibrin polymer. These equations do not influence the amplification of thrombin production and therefore the system (24)–(29) can be studied independently.

Under the assumption of detailed equilibrium, the concentrations of active coagulation factors can be expressed as follows:

$$[Va] = \frac{\hat{k}_1}{h_1} T, \ [VIIIa] = \frac{\hat{k}_2}{h_2} T, \ [XIa] = \frac{\hat{k}_3}{h_3} T,$$

$$[IXa] = \frac{\hat{k}_3 \hat{k}_4}{h_3 h_4} T, \ [Xa] = \frac{\hat{k}_3 \hat{k}_4}{h_3 h_4} T \left( \frac{\hat{k}_5}{h_5} + \frac{k_{55} \hat{k}_2}{h_2 h_5} T \right). \tag{30}$$

We substitute (30) in the equation of thrombin (29):

$$\frac{\partial T}{\partial t} = D\Delta T + (k_3 T + k_4 T^2 + k_5 T^3) P - \sigma T, \tag{31}$$

where

$$k_3 = \frac{\hat{k}_3 \hat{k}_4 \hat{k}_5 \hat{k}_6}{h_3 h_4 h_5}, \quad k_4 = \frac{\hat{k}_2 \hat{k}_3 \hat{k}_4 \hat{k}_{55} \hat{k}_6}{h_2 h_3 h_4 h_5} + \frac{\hat{k}_1 \hat{k}_3 \hat{k}_4 \hat{k}_5 \hat{k}_{66}}{h_1 h_3 h_4 h_5}, \quad k_5 = \frac{\hat{k}_1 \hat{k}_2 \hat{k}_3 \hat{k}_4 \hat{k}_{55} \hat{k}_{66}}{h_1 h_2 h_3 h_4 h_5}.$$

Eq (31) gives a good approximation of the clot growth rate described by the system (24)–(29) [37].

# Appendix B: Model parameters

**Table 1. Numerical values of the parameters used in the model of thrombin generation.**

| Simulation | Value | Unit | Description |
|---|---|---|---|
| $dt$ | 0.02 | $s$ | time step |
| Kinetic rates | Value | Unit | Reaction |
| $a_1$ | $1 \times 10^{-5}$ | $nM^{-2}\,s^{-1}$ | $X$ activation by $TF - VII_a$ |
| $a_2$ | $1 \times 10^{-10}$ | $nM^{-1}\,s^{-1}$ | $X$ activation by $IIa - TF - VII$ |
| $a_3$ | $1.34 \times 10^{-2}$ | $nM^{-1}\,Ls^{-1}$ | $X$ activation by platelets |
| $a_4$ | $8 \times 10^{-6}$ | $nM^{-1}\,s^{-1}$ | $X$ inactivation by antithrombin |
| $a_1$ | $1 \times 10^{-5}$ | $nM^{-1}\,s^{-1}$ | $X$ activation by $TF - VII_a$ |
| $a_1$ | $1 \times 10^{-5}$ | $nM^{-1}\,s^{-1}$ | $X$ activation by $TF - VII_a$ |
| $b_1$ | $3 \times 10^{-2}$ | $nM^{-1}\,s^{-1}$ | $II$ activation by $B_a$ |
| $k_3$ | $1.5 \times 10^{-5}$ | $nM^{-1}\,s^{-1}$ | $II$ activation by $T$ |
| $k_4$ | $8 \times 10^{-6}$ | $nM^{-3}\,s^{-1}$ | $II$ activation by $T^2$ |
| $k_5$ | $1 \times 10^{-10}$ | $nM^{-3}\,s^{-1}$ | $II$ activation by $T^3$ |
| $b_2$ | $1.5 \times 6.7$ | $nM^{-6}\,s^{-1}$ | $IIa$ inactivation by antithrombin |
| Initial concentrations | Value | Unit | Description |
| $TF$ | $0.5 \times 10^{-3}$ | $nM$ | tissue factor concentrations |
| $[VIIa]$ | 10 | $nM$ | factor VIIa concentration |
| $[II]_0$ | 950 | $nM$ | initial prothrombin concentration |
| $[X]_0$ | 80 | $nM$ | initial FX concentration |
| $[ATIII]_0$ | 3000 | $nM$ | $X$ initial antithrombin concentration |

**Table 2. Values of all parameters used in the model of thrombus formation in the flow.**

| Simulation | Value | Unit | Description |
|---|---|---|---|
| $h$ | 0.02 | $mm$ | space step |
| $dt$ | 0.005 | $s$ | time step |
| Diffusion coefficients | Value | Unit | |
| $D$ | $510^{-5}$ | $mm^2/s$ | Blood factors diffusion coefficient [22] |
| $D_p$ | $2.510^{-5}$ | $mm^2/s$ | Platelets diffusion coefficient [22] |
| Kinetic rates | Value | Unit | Reaction |
| $k_1$ | $7 \times 10^{-5}$ | $10^{-9}\,Ls^{-1}$ | $P$ activation by $\phi_c$ (fitted) |
| $k_2$ | $7.5 \times 10^{-6}$ | $nM^{-1}\,s^{-1}$ | $P$ activation by $B_a$ [14] |
| $k_3$ | $1.5 \times 10^{-5}$ | $nM^{-1}\,s^{-1}$ | $P$ activation by $T$ (fitted) |
| $k_4$ | $8 \times 10^{-6}$ | $nM^{-2}\,s^{-1}$ | $P$ activation by $T^2$ (fitted) |
| $k_5$ | $1 \times 10^{-10}$ | $nM^{-3}\,s^{-1}$ | $P$ activation by $T^3$ (fitted) |
| $k_6$ | $4.817 \times 10^{-6}$ | $nM^{-1}\,s^{-1}$ | $T$ inactivation by $A$ [36] |
| $k_7$ | $1 \times 10^{-9}$ | $10^{-9}\,Ls^{-1}$ | $B$ activation by $\phi_c$ (fitted) |
| $k_8$ | $5.2173 \times 10^{-5}$ | $nM^{-1}\,s^{-1}$ | $B$ activation by $T$ (estimated from [37]) |
| $k_9$ | $2.223 \times 10^{-9}$ | $nM^{-1}\,s^{-1}$ | $B_a$ inactivation by $A$ [36] |
| $k_{10}$ | 0.05 | $s^{-1}$ | $F_g$ activation by $T$ [38] |
| $K_{10}$ | 3160 | $nM$ | substrate fibrin concentration [38] |
| $k_{11}$ | 0.1 | $s^{-1}$ | $F \rightarrow F_p$ fibrin polymerization |
| $k_{12}$ | 0.002 | $nM^{-1}\,s^{-1}$ | platelets activation by thrombin (estimated from [35]) |

*(Continued)*

**Table 2.** (Continued)

| | Value | Unit | Description |
|---|---|---|---|
| $k_{13}$ | $4 \times 10^{-9}$ | $10^{-9} \, Ls^{-1}$ | platelets activation by platelets (estimated from [22]) |
| Kinetic constants | Value | Unit | Description |
| $\alpha_1$ | $7.7 \times 10^4$ | $nM^{-1} \, s^{-1}$ | $\alpha_1 = k_f^- k_f T_F^*$ |
| $\beta_1$ | 0.225 | $nM^{-1}$ | $\beta_1 = k_f^+ / k_f^-$ |
| $\phi_{max}$ | 400 | $10^9 \, L^{-1}$ | maximal density of platelets |
| Kinetic concentrations | Value | Unit | Description |
| $P_0$ | 1400 | $nM$ | prothrombin concentration [14] |
| $A_0$ | 3400 | $nM$ | antithrombin concentration [14] |
| $B^0$ | 200 | $nM$ | sum of factors IX and X concentrations [14] |
| $[VII]$ | 10 | $nM$ | factor VII concentration [14] |
| $Fg_0$ | 7000 | $nM$ | fibrinogen concentration [14] |
| $\phi_f^0$ | 300 | $10^9 \, L^{-1}$ | platelets density [14] |
| $\phi_c^0$ | 1 | $10^9 \, L^{-1}$ | activated platelets bound to the subendothelium [14] |
| Flow properties | Value | Physical | Description |
| $\rho$ | $1.06 \times 10^{-6}$ | $kgmm^{-3}$ | plasma density |
| $\nu$ | 1.3 | $mm^{-2} \, s^{-1}$ | blood plasma viscosity |
| $\alpha$ | $6 \times 10^{-4}$ | $mm$ | fiber radius |
| $a$ | $5.6 \times 1^{-3}$ | $s \, mm^{-1}$ | thrombin removal rate by flow |

## Author Contributions

**Conceptualization:** Anass Bouchnita, Patrice Nony, Yuri Vassilevski, Vitaly Volpert.

**Data curation:** Anass Bouchnita, Patrice Nony.

**Formal analysis:** Kirill Terekhov, Patrice Nony, Yuri Vassilevski, Vitaly Volpert.

**Funding acquisition:** Kirill Terekhov, Yuri Vassilevski, Vitaly Volpert.

**Investigation:** Anass Bouchnita, Kirill Terekhov, Yuri Vassilevski, Vitaly Volpert.

**Methodology:** Anass Bouchnita, Kirill Terekhov, Patrice Nony, Yuri Vassilevski.

**Project administration:** Vitaly Volpert.

**Resources:** Kirill Terekhov, Yuri Vassilevski.

**Software:** Kirill Terekhov, Yuri Vassilevski.

**Supervision:** Patrice Nony, Yuri Vassilevski, Vitaly Volpert.

**Validation:** Anass Bouchnita, Kirill Terekhov, Patrice Nony, Yuri Vassilevski, Vitaly Volpert.

**Visualization:** Anass Bouchnita, Kirill Terekhov.

**Writing – original draft:** Anass Bouchnita, Kirill Terekhov.

**Writing – review & editing:** Patrice Nony, Yuri Vassilevski, Vitaly Volpert.

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
