## [Decision Letter · Decision Letter 0]

14 Nov 2019

PONE-D-19-27289

A mathematical model to quantify the effects of platelet count, shear rate, and injury size on the initiation of blood coagulation under venous flow conditions

PLOS ONE

Dear Dr. Bouchnita,

Thank you for submitting your manuscript to PLOS ONE. After careful consideration, we feel that it has merit but does not fully meet PLOS ONE’s publication criteria as it currently stands. Therefore, we invite you to submit a revised version of the manuscript that addresses the points raised during the review process.

We would appreciate receiving your revised manuscript by Dec 29 2019 11:59PM. To enhance the reproducibility of your results, we recommend that if applicable you deposit your laboratory protocols in protocols.io, where a protocol can be assigned its own identifier (DOI) such that it can be cited independently in the future. For instructions see: http://journals.plos.org/plosone/s/submission-guidelines#loc-laboratory-protocols

We look forward to receiving your revised manuscript.

Kind regards,

Fang-Bao Tian

Academic Editor

PLOS ONE

Journal Requirements:

Reviewers' comments:

Reviewer's Responses to Questions

**Comments to the Author**

1. Is the manuscript technically sound, and do the data support the conclusions?

Reviewer #1: Partly

Reviewer #2: Yes

2. Has the statistical analysis been performed appropriately and rigorously? 

Reviewer #1: N/A

Reviewer #2: Yes

3. Have the authors made all data underlying the findings in their manuscript fully available?

Reviewer #1: Yes

Reviewer #2: Yes

4. Is the manuscript presented in an intelligible fashion and written in standard English?

Reviewer #1: Yes

Reviewer #2: Yes

5. Review Comments to the Author

Reviewer #1: The manuscript proposes a model describing the formation of the thrombus in venous vessel. The model is extended from the previous work of the authors, by adding the participation of the platelets. To identify the parameters, the authors adopt another so-called thrombin generation model which does not include the advection of the flow. This gives the value of the parameters by fitting the output data to the experimental data. Using the fitted parameters, their model can reproduce the threshold response to the shear rate in clotting initiation. Further more, they deviate a simplified 1D equation from their model, which could also capture the threshold feature. The simplified model may provide a cheap way to estimate the influence of the various parameters on the thrombus formation.

Major comments:

1) The control parameter, shear rate, is implied by a constant pressure in this work. However, the flow will change with the growth of the thrombus in the presence of constant pressure boundary. Thus the shear rate is not a well controlled parameter in the simulations. While the compared experiment used the constant velocity or flux rate as the imposed flow.

2) The system is symmetrical. But the asymmetrical colour map appears in Fig. 4 and Fig. 5. Could the authors explain the reason?

3) In Fig. 6 a, the value of platelet count is not given for the platelets rich plasma case. I guess that it is 150 \\times 10^9 L^-1. This gives the threshold shear rate at about 80 s^-1. While the experiment in ref 10 used the value of platelet count as 300 \\times 10^9 L^-1 and gave the threshold shear rate at about 80 s^-1. Speculated from Fig. 7, the threshold shear rate for 300 \\times 10^9 L^-1 platelet count would be around 160 s^-1 in the simulations. Could the authors explain this deviation between the experimental observation and the simulation.

4) As the authors indicate that the length of the tube will significantly influence the clotting initiation time, could the author do more discussion about the effect of the tube length and the tube diameter as well?

Minor comments:

1) The quality of the figures should be improved. For example, the colour bar can not been seen, the legend in Fig. 3 a miss \\times, the label of sub-figures sometimes is capital while sometimes not ...

2) Some typo was found. Such as 'thrombin [II]' below the equation 18. I think this should be 'prothrombin [II]'. Please do a careful checking, since the model contains a lot of parameters, small typo may make the reader lost.

Reviewer #2: Comments on research paper titled: "A mathematical model to quantify the effects of platelet

count, shear rate, and injury size on the initiation of blood coagulation under venous flow conditions" by Bouchnita. A et al.

This work presents a quantitative study of the effects of platelet count, wall shear rate,

and injury size on the initiation of blood coagulation based on the analysis of the model

and on numerical simulations. First, they described the mathematical modelling of venous thrombus

growth under flow and derived a simplified model of thrombin distribution. Then, through numerical simulations,

they studied the relationship between platelet count and blood coagulability, the shear rate threshold dependence

on the density of platelets, and the effect of TF surface area on the initiation of blood clotting. The results in this

work should be of interest to PLOS ONE community. However, I think this paper still has some issues to be addressed before

it can be accepted.

1): As I know, the spatiotemporal model of thrombus development and thrombin generation model have been previously proposed.

What's the difference between them and models presented in this work?

2): Authors simplified the thrombin distribution with 1D model, however, this is a 3D problem. If it is appropriate to assume

the distribution of thrombin in x-z plane (using the coordinate in the work)?

Minor:

1): In the introduction, author should clarify why the platelet play different roles in arterial and venous flow.

2): In the fig 3. The results have good agreement with the experiments. However, the simulation exlcudes the effects of RBCs

in the venous flow. In the Discussion, author states that RBCs will accelerate the margination of platelets, if it is possilbe

to include the RBCs effect in the model.

3): The occlusion of vessel is presented in fig 6. How to estimate the occlusion, please clarify this.

4): Author just use two cases (different platelet count) in microfluidc capillary. This is not enough to make comparison.

6. PLOS authors have the option to publish the peer review history of their article (what does this mean?). If published, this will include your full peer review and any attached files.

Reviewer #1: No

Reviewer #2: No

---

## [Author Response · Author response to Decision Letter 0]

13 May 2020

A point-by-point response to the reviewer's comments is provided with the revised manuscript. A color-marked copy of the manuscript is also uploaded.

---

## [Decision Letter · Decision Letter 1]

25 May 2020

PONE-D-19-27289R1

A mathematical model to quantify the effects of platelet count, shear rate, and injury size on the initiation of blood coagulation under venous flow conditions

PLOS ONE

Dear Dr. Bouchnita,

Thank you for submitting your manuscript to PLOS ONE. After careful consideration, we feel that it has merit but does not fully meet PLOS ONE’s publication criteria as it currently stands. Therefore, we invite you to submit a revised version of the manuscript that addresses the points raised during the review process.

We look forward to receiving your revised manuscript.

Kind regards,

Fang-Bao Tian

Academic Editor

PLOS ONE

Reviewers' comments:

Reviewer's Responses to Questions

**Comments to the Author**

1. If the authors have adequately addressed your comments raised in a previous round of review and you feel that this manuscript is now acceptable for publication, you may indicate that here to bypass the “Comments to the Author” section, enter your conflict of interest statement in the “Confidential to Editor” section, and submit your "Accept" recommendation.

Reviewer #1: (No Response)

Reviewer #2: All comments have been addressed

2. Is the manuscript technically sound, and do the data support the conclusions?

Reviewer #1: Partly

Reviewer #2: Yes

3. Has the statistical analysis been performed appropriately and rigorously? 

Reviewer #1: N/A

Reviewer #2: Yes

4. Have the authors made all data underlying the findings in their manuscript fully available?

Reviewer #1: Yes

Reviewer #2: Yes

5. Is the manuscript presented in an intelligible fashion and written in standard English?

Reviewer #1: Yes

Reviewer #2: Yes

6. Review Comments to the Author

Reviewer #1: The authors have addressed my comments. However, the answer for the second question can not convince the referee. The Fig.5 B is especially unreasonable. There probably be some technical mistakes. This might be a small problem, which may not influence the other results and conclusions, but a wrong figure is obviously unacceptable. I think this paper could not be accepted unless the authors address this problem. Another minor comment is that the figures are made negligently, such as the unreadable color map in Fig.4 and Fig.5, the legend in Fig.3 a) 200 10^9, the size of the text in all the figures is careless, especially Fig.8 B) ...

Reviewer #2: The authors already solved all the aspects I raised in the review stage, therefore I recommend the publication.

7. PLOS authors have the option to publish the peer review history of their article (what does this mean?). If published, this will include your full peer review and any attached files.

Reviewer #1: No

Reviewer #2: No

---

## [Author Response · Author response to Decision Letter 1]

10 Jun 2020

We Reviewer #1 for their further attention to our manuscript. Answers to the reviewer's comments are provided in the response letter.

---

## [Editor Report · Decision Letter 2]

16 Jun 2020

A mathematical model to quantify the effects of platelet count, shear rate, and injury size on the initiation of blood coagulation under venous flow conditions

PONE-D-19-27289R2

Dear Dr. Bouchnita,

We’re pleased to inform you that your manuscript has been judged scientifically suitable for publication and will be formally accepted for publication once it meets all outstanding technical requirements.

Kind regards,

Fang-Bao Tian

Academic Editor

PLOS ONE
---

## [Editor Report · Acceptance letter]

1 Jul 2020

PONE-D-19-27289R2 

A mathematical model to quantify the effects of platelet count, shear rate, and injury size on the initiation of blood coagulation under venous flow conditions 

Dear Dr. Bouchnita:

I'm pleased to inform you that your manuscript has been deemed suitable for publication in PLOS ONE. Congratulations! Your manuscript is now with our production department. 

Kind regards, 

on behalf of

Dr. Fang-Bao Tian 

Academic Editor

PLOS ONE